# Immune Control and Vaccination against the Epstein–Barr Virus in Humanized Mice

**DOI:** 10.3390/vaccines7040217

**Published:** 2019-12-17

**Authors:** Christian Münz

**Affiliations:** Viral Immunobiology, Institute of Experimental Immunology, University of Zürich, Winterthurerstrasse 190, CH-8057 Zürich, Switzerland; christian.muenz@uzh.ch

**Keywords:** lymphoma, infectious mononucleosis, CD8^+^ T cells, NK cells, virus-like particles

## Abstract

Mice with reconstituted human immune system components (humanized mice) offer the unique opportunity to test vaccines preclinically in the context of vaccine adjuvant sensing by human antigen presenting cells and priming of human cytotoxic lymphocyte populations. These features are particularly attractive for immune control of the Epstein–Barr virus (EBV), which represents the most potent growth-transforming pathogen in man and exclusively relies on cytotoxic lymphocytes for its asymptomatic persistence in the vast majority of healthy virus carriers. This immune control is particularly impressive because EBV infects more than 95% of the human adult population and persists without pathology for more than 50 years in most of them. This review will discuss the pathologies that EBV elicits in humanized mice, which immune responses control it in this model, as well as which passive and active vaccination schemes with adoptive T cell transfer and with virus-like particles or individual antigens, respectively, have been explored in this model so far. EBV-specific CD8^+^ T cell priming in humanized mice could provide crucial insights into how cytotoxic lymphocytes against other viruses and tumors might be elicited by vaccination in humans.

## 1. Introduction on the Epstein Barr Virus

The Epstein–Barr virus (EBV) is a common human γ-herpesvirus that persistently infects more than 95% of the human adult population [1]. Humans are the virus’ only reservoir and its life-cycle is perfectly adapted to human B cell biology, utilizing the differentiation stages of this lymphocyte population to persist in long-lived memory cells without viral protein expression (latency 0) and to reactivate for viral particle production and transmission upon plasma cell differentiation [2]. However, in order to reach these terminal differentiation stages of human B cells, EBV risks B cell transformation, both via viral oncogene expression from latent EBV open reading frames (ORFs) [3] and via induction of the somatic hypermutation machinery of B cells that is designed to improve the B cell receptor, but also responsible for oncogenic mutations such as c-myc translocation in Burkitt lymphoma [4,5]. EBV is even the only virus to date that can readily transform its main host cell, the human B cell, into immortalized lymphoblastoid cell lines (LCLs) in cell culture. Accordingly, it is associated with several B cell lymphomas, including Burkitt lymphoma, Hodgkin lymphoma, diffuse large B cell lymphomas (DLBCL) and post-transplant lymphoproliferative disease (PTLD) [6]. These can express all latent EBV gene products, including the six EBV nuclear antigens (EBNAs), the two latent membrane proteins (LMPs), more than 40 miRNAs, the two EBV encoded small RNAs (EBERs), as so-called latency III in tumors like PTLD and in LCLs, as well as naïve tonsillar B cells in healthy EBV carriers [1]. Alternatively, smaller sets of these with EBNA1 as the sole viral protein are present in latency I that is found in Burkitt lymphoma and infected homeostatically proliferated memory B cells, or EBNA1 plus the two LMPs in latency II in Hodgkin lymphoma and germinal center B cells of healthy virus carriers [1]. The reduction in viral protein expression renders the respective tumors and B cell differentiation stages less immunogenic, and even if all of these malignancies increase upon immune suppression by human immunodeficiency virus (HIV) co-infection, latency III lymphomas are only observed after nearly complete immune deficiency [7]. In addition to these B cell lymphomas, EBV is associated with natural killer (NK)/T cell lymphomas, epithelial cell cancers such as nasopharyngeal carcinoma, and rare smooth muscle cancers in severely immune compromised patients [6]. It remains, however, poorly understood under which circumstances EBV contributes to the transformation of these non-B cells. Nevertheless, especially EBV-associated epithelial carcinomas contribute largely to the around 200,000 new tumors that emerge every year and are associated with this tumor virus [8]. Overall, EBV is estimated to be associated with around 1%–2% of all malignancies in humans [9,10], and this alone justifies the development of vaccination against this pathogen.

In addition, EBV is associated with a number of immune pathologies [11]. These include the symptomatic primary infection infectious mononucleosis (IM) [12], hemophagocytic lymphohistiocytosis (HLH) [13], and possibly the autoimmune disease multiple sclerosis (MS) [14]. All of these result from hyperactivation of cytokine producing T cells that might not sufficiently control EBV via cytotoxicity [15,16]. Especially IM is quite frequent in Northern latitudes and affects primarily adolescents upon delayed primary EBV infection after the first decade of life [12]. In this EBV seronegative population, 30%–50% can experience primary infection with IM [17,18], affecting around 10% of the overall population. This high frequency makes prevention of IM an attractive vaccination goal for initial clinical trials [19,20].

## 2. Modelling Tumorigenesis and Immune Pathology by EBV in Humanized Mice

The lack of orthologues for EBV in rodents [21] complicates modelling of its pathologies in preclinical animal models. Therefore, other laboratories and ours started to explore EBV infection in mice with human immune system components that were reconstituted from CD34^+^ hematopoietic progenitor cells after perinatal intrahepatic injection (humanized mice) [22,23,24,25,26,27] (Figure 1). EBV persists in this model in human B cells and can even access latency 0 [28]. With respect to EBV-associated diseases, latency III B cell lymphomas can be observed that grow as LCLs in culture after ex vivo isolation [29]. These are heavily infiltrated with T cells that depend on EBNA3B for the induction of chemokines that attract them into the tumor microenvironment. EBNA3B deficient EBV infection results in increased formation of tumors that transcriptionally resemble DLBCL and indeed mutations in this latency gene product have been detected in a subset of patients that suffer from this tumor [29,30]. EBV also contributes to 90% of primary effusion lymphomas (PEL), which are in all cases co-infected with its closest relative among human viruses, the Kaposi sarcoma-associated herpesvirus (KSHV) [31]. This tumor has a characteristic plasma cell differentiation [32]. Double-infection of EBV with KSHV increases lymphomagenesis in humanized mice [33], at the same time as it supports KSHV persistence. The resulting tumors display plasma cell differentiation and increased lytic EBV reactivation, reminiscent of PELs. Indeed, early, but presumably not late lytic EBV protein expression supports lymphomagenesis [24,34,35]. Possibly, due to the short duration of the experimental EBV infection in humanized mice of so far only up to three months, no additional somatic mutations were acquired to develop latency I and II tumors [36]. However, CD4^+^ T cell-dependent transcripts of these latencies have been detected during EBV infection of humanized mice [37], suggesting germinal center dependence of these lower latencies and constituting premalignant states that could eventually develop into Burkitt or Hodgkin lymphomas.

In addition to these malignancies, humanized mice also develop EBV-induced immunopathologies. Infection with 10^5^, but not 10^3^ viral particles leads to massive CD8^+^ T cell expansion with similar kinetics and immunopathology as during IM [38]. During IM in adolescents a large fraction of these expanding CD8^+^ T cells are directed against lytic EBV antigens [39,40]. Similarly, half of the CD8^+^ T cell expansion during IM-like disease in humanized mice is lost after infection with a BZLF1-deficient recombinant EBV virus that is no longer able to switch into lytic infection [41]. With even higher cytokine production during EBV infection of humanized mice, symptoms of HLH can also be observed [42,43], most likely resulting from myeloid cell activation by T cell produced cytokines. However, MS-like symptoms have so far not been observed during EBV infection of humanized mice, even if human CD4^+^ T cells can efficiently home to the mouse’s CNS under inflammatory conditions in humanized mice [44,45]. Thus, humanized mice develop several pathologies that are associated with EBV. Others that are not yet observed might be elicited with recombinant viruses that favor their associated EBV infection programs.

## 3. Characteristics of Cell-Mediated Immune Control against EBV in Humanized Mice

Fortunately, such EBV-associated pathologies rarely occur in healthy EBV carriers despite the long duration of persistent infection with nearly all individuals getting infected with EBV before 2 years of age in Sub-Saharan Africa and two thirds acquiring the virus at a similar young age in Europe and North America [12]. This represents one of the most formidable feats of our immune system, controlling the most potent transforming virus for more than 50 years. The cornerstones of this near perfect immune control are known through therapeutic interventions against EBV-associated malignancies and primary immunodeficiencies that predispose for virus-associated pathologies [15,16,46]. These studies identify cytotoxic lymphocytes, primarily CD8^+^ T cells, as the main component of EBV-specific immune control. In contrast, antibody responses and type I as well as type II interferon (IFN) production are not required, and might in the case of the type II IFN-γ rather elicit immunopathologies like HLH in the absence of sufficient cytotoxicity [15]. Accordingly, type I IFNs and their main producers, the plasmacytoid dendritic cells (pDCs), are not required for EBV-specific immune control in humanized mice [47].

Similar to patients, cytotoxic lymphocytes, CD8^+^ T, and NK cells respond to EBV infection in humanized mice with activation and expansion [36]. In contrast to these cytotoxic lymphocyte responses, humanized mice without additional modifications, like thymic stromal lymphopoietin (TSLP) overexpression [48], develop only IgM antibody responses during EBV infection, but not class-switched and affinity-matured humoral immune responses [36]. The expanding cytotoxic T and NK lymphocyte populations control EBV infection in humanized mice, because their antibody-mediated depletion increases viral loads and associated lymphoma formation [26,49,50,51,52]. The respective CD8^+^ T cell populations balance inhibitory with activating co-receptor expression and retain superior cytotoxicity despite expression of the inhibitory PD-1 molecule [38]. The activating co-receptor 2B4 is required for EBV-specific immune control by CD8^+^ T cells in humanized mice [53]. Furthermore, they maintain homing to secondary lymphoid organs and even germinal centers (CXCR5^+^) where EBV-infected B cells can be found [38,54]. CD8^+^ T cell expansion is directed towards both latent and lytic EBV antigens and adoptive transfer of clonal CD8^+^ T cells that are specific for the early lytic antigen BMLF1 can decrease lytic replication [41]. Furthermore, a T cell receptor (TCR)-like antibody recognizing a LMP2 peptide presented on the MHC class I molecule HLA-A2 was isolated [55]. This antibody, formulated as a T cell engager with an additional specificity for CD3, was able to redirect adoptively transferred T cells towards LCL elimination in immune compromised mice. Moreover, LMP1-specific HLA-A2 restricted TCRs confer efficient T cell recognition of EBV-associated tumor cells in vitro [56]. These studies confirm that CD8^+^ T cell responses against LMP1 and LMP2 are efficient in targeting EBV-associated lymphomas, as has also been observed in the clinic [57]. In addition, CD8^+^ T cells against an early lytic EBV antigen could be therapeutically explored.

Lytic EBV replication is also suppressed by NK cells in humanized mice, resulting in increased tumor formation, viral loads, and IM-like CD8^+^ T cell expansion upon their antibody-mediated depletion [50,51]. Particularly early differentiated NKG2A^+^killer immunogblobulin-like receptor (KIR)^−^ NK cells that constitute the majority of the human NK cell compartment early in life, expand during EBV infection and kill lytic EBV replicating cells efficiently [18,58,59]. In addition to NK cells, NKT and γδ T cells as additional innate lymphocyte populations can suppress EBV infection in humanized mice [60,61,62]. Thus, innate in addition to adaptive cytotoxic lymphocytes might contribute to immune control of EBV, and their contributions can be analyzed in humanized mice.

Finally, cytotoxic CD4^+^ T cells have previously been described to efficiently target EBV infected lymphoma cells [63]. In particular, EBNA1 and late lytic EBV antigens have been explored to target EBV-transformed B cells [64,65,66,67,68]. Late lytic EBV antigen-specific CD4^+^ T cells can indeed also restrict EBV infection in humanized mice [69]. However, their limited expansion during primary infection [70] and absence of EBV-associated pathologies in patients with MHC class II deficiencies [71] argues against CD4^+^ T cells being sufficient for EBV-specific immune controls. Thus, primarily CD8^+^ T and NK cells have been identified as sources of EBV-specific immune control in humanized mice.

## 4. Adoptive EBV-Specific T Cell Transfer in Humanized Mice

EBV associated lymphomas are one of the first tumor entities for which adoptive transfer of virus-specific T cell lines into patients were explored [72,73]. While these were originally expanded from peripheral blood mononuclear cells (PBMCs) with autologous LCLs, mixtures of defined HLA matched EBV-specific T cell clones [74] and T cells, stimulated or isolated for EBNA1, LMP1, and LMP2 recognition, were explored more recently with clinical success [75,76]. LCL-expanded T cell lines have been tested in immune-compromised mice to eradicate subcutaneous and intraperitoneal LCL tumors [29,77]. In addition, EBV-specific T cell clones have been transferred into EBV-infected humanized mice or LCL-transplanted immune-compromised mice [41,69] (Figure 1). These included CD8^+^ T cell clones specific for LMP2 and the early lytic EBV antigen BMLF1, or CD4^+^ T cell clones specific for EBNA1, EBNA3B, EBNA3C, as well as the late lytic EBV antigens BNRF1 (major tegument protein) and BLLF1 (envelope gp350 protein). Of these, only BMLF1-specific CD8^+^ T cells suppressed lytic EBV replication, and BLLF1-specific CD4^+^ T cells delayed LCL growth at doses of adoptively transferred T cells of 10^6^ and 10^7^, respectively [41,69]. The TCRs of two BMLF1- and LMP2-specific CD8^+^ T cell clones were cloned into retroviral expression vectors to express them in splenocytes prior to adoptive transfer into autologously reconstituted humanized mice [38]. At least after adoptive transfer of 200,000 TCR transgenic T cells, no alteration of EBV loads were detected. However, LMP2-specific TCR transgenic CD8^+^ T cells expanded 20-fold, while only an around 5-fold expansion was detected for CD8^+^ T cells carrying the BMLF1-specific TCR [38]. Therefore, distinct EBV-specific T cells can be explored in humanized mice for their activation, differentiation, and tissue distribution. In order for the adoptive T cell transfer to affect EBV loads and virus-associated lymphomagenesis, however, probably in excess of 10^6^ specific T cells need to be transferred in this model.

## 5. Vaccination against EBV in Humanized Mice

In part this lack of increased immune control of EBV upon adoptive transfer of viral antigen-specific T cell populations is due to priming of protective T cell responses by the infection itself [26,49,52,53]. One should be able to harness this endogenous T cell response in humanized mice by efficient vaccine formulations to control successive EBV infection, but this has not been easy so far, in part because we lack efficient vaccine formulations that prime protective CD8^+^ T cell responses [78]. Humanized mice are particularly attractive to explore CD8^+^ T cell priming because they can sense vaccine adjuvants for the activation of immune responses similar to humans without the bias towards mycobacterial product detection, e.g., complete Freund’s adjuvant (CFA), by toll-like receptor 4 (TLR4) that is inherent to mice [67]. However, even with human-biased adjuvants like the TLR3 agonist poly I:C and individual EBV antigens, such as EBNA1, it has been difficult to elicit more than low frequency CD4^+^ T cell responses after vaccination [67,79] (Table 1). In these vaccine formulations the EBV antigens were targeted to endocytosis receptors on dendritic cells (DCs), such as the decalectin DEC-205 [67,79] (Figure 1). Unfortunately, this antigen delivery, even when various endocytic receptors on DCs are compared, favors primarily antigen processing onto MHC class II molecules [80]. This predominance of DEC-205-targeted antigen presentation on MHC class II molecules is even observed for mouse DEC-205 in inbred laboratory mice [81]. Furthermore, this might be the wrong antigen presenting cell population to target, because we and others have not found any indication of conventional DC activation during primary EBV infection [18,47] and removing the main viral component of conventional DC activation, EBERs [82], does not alter T cell priming during EBV infection in humanized mice [83]. Accordingly, primary immunodeficiencies that affect IL-12 as the hallmark cytokine of conventional DC activation and its signaling for IFN-γ induction do not confer susceptibility to EBV-induced pathogenesis [15]. Thus, conventional DCs do not seem to sufficiently sense EBV infection to participate in its T cell priming.

Plasmacytoid DCs, on the other hand, recognize EBV and seem to be depleted from peripheral blood during primary EBV infection [18,47,85,86,87]. However, their presence and type I IFN production is dispensable for T cell priming in humanized mice [47], and also primary immunodeficiencies that affect the type I IFN pathway do not confer enhanced susceptibility to EBV [15]. Therefore, vaccine targeting to both conventional and plasmacytoid DCs might not be necessary to induce protective T cell responses to EBV.

Instead, EBV-infected B cells might play a large part in priming and expanding CD8^+^ T cell responses that control this human tumor virus [81]. Indeed, the only vaccine formulation that has so far shown protection from a challenge by EBV infection in humanized mice uses the EBV tropism for B cells [84]. The respective virus-like particles (VLPs) represent EBV without its DNA genome. However, while these VLPs alone did not induce protection from high viral loads upon EBV infection, inclusion of EBNA1 into the tegument of these VLPs was able to induce protective cytotoxic CD4^+^ T cell responses (Figure 1 and Table 1). This suggests that latent EBV antigens targeting B cells might elicit higher cytotoxic CD4^+^ T cell responses than when targeting DCs [67,79], and should be further explored for EBV-specific vaccination. However, the EBNA1 containing VLPs do only induce CD4^+^ T cell responses, while cytotoxic EBV-specific CD8^+^ T cell responses are the cornerstone of natural immune control of this human tumor virus. Along these lines live attenuated viruses still remain the gold standard to deliver antigens into the cytosol for CD8^+^ T cell stimulation [78]. Especially adenoviruses seem to be excellent vectors to elicit EBNA1-specific CD8^+^ T cell responses, and have in combination with DEC-205 targeting of EBNA1 or modified vaccinia virus Ankara (MVA) expressing EBNA1 induced long-lived CD8^+^ T cell responses in human DEC-205 transgenic mice that protected from EBNA1 expressing a syngeneic lymphoma challenge [80]. These viruses and possibly even an attenuated and highly immunogenic EBV virus should be further explored for vaccine-induced priming of EBV-specific immune control by CD8^+^ T cells.

## 6. Conclusions and Outlook

Among human pathogens that are associated with tumor formation, efficient vaccines against the human papilloma virus (HPV) and hepatitis B virus (HBV), as well as antibiotic treatment against *Helicobacter pylori* and protease inhibitors against hepatitis C virus (HCV) have been developed [88,89,90]. These represent the infectious disease agents most frequently associated with malignancies in humans [9,10]. EBV is the next most frequent agent that is associated with tumors [8]. Therefore, the time seems ripe to develop an EBV vaccine. The challenge is the requirement for a particular cell-mediated protective immunity against this tumor virus, which seems to rely mainly on cytotoxic CD8^+^ T cell lymphocytes. This challenge could, however, also be a chance to use EBV as a model to develop vaccination schemes that would for the first time allow the efficient priming of protective CD8^+^ T cell responses, including those against other viruses and cancers.

Humanized mice might be an ideal platform for this task, because these models can develop cell-mediated immune control, for example against EBV, but usually fail to raise antibody responses [36]. Furthermore, they respond to vaccines with adjuvant recognition similar to humans [67,91] (Figure 1). However, only the future will tell if these preclinical models will predict vaccine outcomes in humans more efficiently than inbred laboratory mice.

## Figures and Tables

**Figure 1 vaccines-07-00217-f001:**
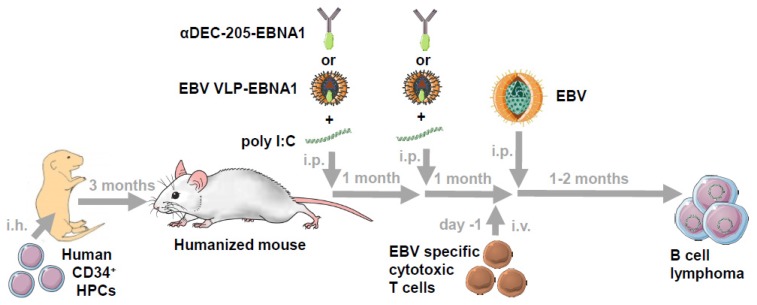
Epstein–Barr virus (EBV)-specific vaccination in humanized mice. Alymphoid mice are perinatally injected with human CD34^+^ hematopoietic progenitor cells (HPCs) via the intrahepatic (i.h.) route. After three months of reconstitution, the human immune compartment composition is determined from peripheral blood. For vaccination experiments these humanized mice were intraperitoneally (i.p.) injected either with virus-like particles (VLPs) or EBV antigen targeted to the endocytic decalectin DEC-205 receptor on dendritic cells plus the TLR3 agonist poly I:C as adjuvant. A homologous boost was given after one month. One month later the vaccinated mice were challenged with intraperitoneal EBV infection. One to two months later viral loads and lymphoma formation was assessed. Only when EBNA1 was included into the VLPs, protection from successive EBV infection was observed. Alternatively, just prior to EBV infection (Day 1), EBV-specific cytotoxic T cells were injected intravenously to monitor EBV specific T cell expansion during infection and to influence the latent or lytic EBV life cycle.

**Table 1 vaccines-07-00217-t001:** Active EBV-specific vaccination approaches that have been tested in humanized mice.

Vaccine	Elicited Immune Response	Effect on EBV Challenge	References
αDEC-205-EBNA1	Low level EBNA1 specific CD4^+^ T cell responses	No protection	[67,79]
EBV VLP	Late lytic antigen specific CD4^+^ T cell responses	No protection	[84]
EBV VLP with EBNA1 in tegument	Late lytic antigen and EBNA1 specific CD4^+^ T cell responses	Protection	[84]

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
