# Peer review of "Immune Control and Vaccination against the Epstein–Barr Virus in Humanized Mice"

_vaccines, 2019, doi:10.3390/vaccines7040217_

Round 1

Reviewer 1 Report

Summary:

This article reviews the utility of humanized mice in modeling tumorigenesis and diseases caused by Epstein-Barr virus (EBV) and discusses the use of humanized mice in developing vaccines against EBV. After an introduction to EBV and its associated malignancies in humans, the establishment of a humanized mice model for EBV infection is described. The pathologies and immune response that develop in the humanized mice model is described and compared to the response in humans. Then, the study of EBV-specific T cells in treated EBV-associated lymphomas is compared in humans and humanized mice along with the viral proteins, both latent and lytic, that have been targeted. Finally, strategies for development of vaccines against EBV using humanized mice are proposed by contrasting effective and ineffective immune responses. Overall, the case is made that humanized mice are a key tool for developing an EBV vaccine.

Broad comments

This is a brief, focused, and up-to-date review article that is thoroughly-referenced. It blends knowledge of the immune response and EBV. In addition to summarizing published data, the author points out the current data suggest strategies that are more promising than others and advocates for vaccine development.

Overall many sentences are long and would be clearer if broken up and/or commas added. Are there drawbacks or known limitations of humanized mice? A table summarizing the experiments using humanized mice to develop an EBV vaccine (section 5) would be helpful.

Specific comments

Line 27 Provide a brief description of stages of EBV latency to define “latency 0”
Lines 37-41 This section needs clarification of which latency type corresponds to which lymphoma.
Line 89 DEC-205 needs to be explained in the figure legend
Line 252 and 254 Capitalize AID

Author Response

Summary:

This article reviews the utility of humanized mice in modeling tumorigenesis and diseases caused by Epstein-Barr virus (EBV) and discusses the use of humanized mice in developing vaccines against EBV. After an introduction to EBV and its associated malignancies in humans, the establishment of a humanized mice model for EBV infection is described. The pathologies and immune response that develop in the humanized mice model is described and compared to the response in humans. Then, the study of EBV-specific T cells in treated EBV-associated lymphomas is compared in humans and humanized mice along with the viral proteins, both latent and lytic, that have been targeted. Finally, strategies for development of vaccines against EBV using humanized mice are proposed by contrasting effective and ineffective immune responses. Overall, the case is made that humanized mice are a key tool for developing an EBV vaccine.

Broad comments

This is a brief, focused, and up-to-date review article that is thoroughly-referenced. It blends knowledge of the immune response and EBV. In addition to summarizing published data, the author points out the current data suggest strategies that are more promising than others and advocates for vaccine development.

Overall many sentences are long and would be clearer if broken up and/or commas added. Are there drawbacks or known limitations of humanized mice? A table summarizing the experiments using humanized mice to develop an EBV vaccine (section 5) would be helpful.

I have now broken up some sentences to clarify their content. Furthermore, I have added on page 4 of the revised manuscript a sentence on the paucity of antibody responses that can be elicited in humanized mice, even so their role in natural immunity to EBV is probably minimal. Finally, I have added table 1 on page 6 of the revised manuscript to summarize the few explored active vaccination approaches against EBV in humanized mice.

Specific comments

Line 27 Provide a brief description of stages of EBV latency to define “latency 0”

Lines 37-41 This section needs clarification of which latency type corresponds to which lymphoma.

I have now clarified the different latency stages in tumors and B cell differentiation stages of healthy virus carriers in lines 38 to 45 of the revised manuscript.

Line 89 DEC-205 needs to be explained in the figure legend

I have now clarified DEC-205 in the figure legend.

Line 252 and 254 Capitalize AID

 I have now capitalized AID in the references.

Reviewer 2 Report

This review article summarizes the current contribution and the expected benefit of the humanized mouse model to better understand the immune response to EBV in human organisms and to develop effective vaccines against this virus. It is well written, comprehensive and concise.

I have only two minor remarks.

Line 46, in the section 1 (Introduction on the Epstein-Barr virus).

Smooth muscle cancers related to EBV are mentioned in the same list as nasopharyngeal carcinomas suggesting that they are at the same level of incidence. This might be misleading for people who are new in the field. EBV-positive leïomyosarcomas are observed in some patients with severe immune deficiencies but one need to acknowledge that they are very rare! In contrast, nasopharyngeal carcinoma is a major public health problem in several countries.

Lines 218 to 222, in the section 5 (Vaccination against EBV in humanized mice).

It seems to me that this sentence is too long and too complicated. May be, it would be useful to split it in 2 or 3 sentences. I think that the term “regular mice” is not precise enough. The corresponding publication deals with a syngenic tumor model in transgenic mice.

Author Response

This review article summarizes the current contribution and the expected benefit of the humanized mouse model to better understand the immune response to EBV in human organisms and to develop effective vaccines against this virus. It is well written, comprehensive and concise.

I have only two minor remarks.

Line 46, in the section 1 (Introduction on the Epstein-Barr virus).

Smooth muscle cancers related to EBV are mentioned in the same list as nasopharyngeal carcinomas suggesting that they are at the same level of incidence. This might be misleading for people who are new in the field. EBV-positive leïomyosarcomas are observed in some patients with severe immune deficiencies but one need to acknowledge that they are very rare! In contrast, nasopharyngeal carcinoma is a major public health problem in several countries.

I have now clarified that EBV associated smooth muscle tumors are rare and occur in severely immune compromised patients.

Lines 218 to 222, in the section 5 (Vaccination against EBV in humanized mice).

It seems to me that this sentence is too long and too complicated. May be, it would be useful to split it in 2 or 3 sentences. I think that the term “regular mice” is not precise enough. The corresponding publication deals with a syngenic tumor model in transgenic mice.

I have now broken up this sentence and clarified that inbred laboratory mice or human DEC-205 transgenic mice with syngeneic lymphoma models were used. The revised text passages can be found on pages 5, 6 or 7.

Reviewer 3 Report

The article is nicely written with sufficient literature citations and discussions. The english language at few places, was confusing and may need through editing to correct all grammatical issues in the article. 

Author Response

The article is nicely written with sufficient literature citations and discussions. The English language at few places, was confusing and may need through editing to correct all grammatical issues in the article.

I have broken up some sentences and clarified them with additional descriptions.